# Manipulation of Spray-Drying Conditions to Develop an Inhalable Ivermectin Dry Powder

**DOI:** 10.3390/pharmaceutics14071432

**Published:** 2022-07-08

**Authors:** Tushar Saha, Shubhra Sinha, Rhodri Harfoot, Miguel E. Quiñones-Mateu, Shyamal C. Das

**Affiliations:** 1School of Pharmacy, University of Otago, Dunedin 9054, New Zealand; tushar.saha@postgrad.otago.ac.nz; 2Department of Microbiology and Immunology, School of Biomedical Sciences, University of Otago, Dunedin 9054, New Zealand; shubhra.sinha@otago.ac.nz (S.S.); rhodri.harfoot@otago.ac.nz (R.H.); miguel.quinones-mateu@otago.ac.nz (M.E.Q.-M.); 3Webster Centre for Infectious Diseases, University of Otago, Dunedin 9054, New Zealand

**Keywords:** respiratory disease, inhalable dry powder, antiviral treatment, ivermectin

## Abstract

SARS-CoV-2, the causative agent of COVID-19, predominantly affects the respiratory tract. As a consequence, it seems intuitive to develop antiviral agents capable of targeting the virus right on its main anatomical site of replication. Ivermectin, a U.S. FDA-approved anti-parasitic drug, was originally shown to inhibit SARS-CoV-2 replication in vitro, albeit at relatively high concentrations, which is difficult to achieve in the lung. In this study, we tested the spray-drying conditions to develop an inhalable dry powder formulation that could ensure sufficient antiviral drug concentrations, which are difficult to achieve in the lungs based on the oral dosage used in clinical trials. Here, by using ivermectin as a proof-of-concept, we evaluated spray-drying conditions that could lead to the development of antivirals in an inhalable dry powder formulation, which could then be used to ensure sufficient drug concentrations in the lung. Thus, we used ivermectin in proof-of-principle experiments to evaluate our system, including physical characterization and in vitro aerosolization of prepared dry powder. The ivermectin dry powder was prepared with a mini spray-dryer (Buchi B-290), using a 2^3^ factorial design and manipulating spray-drying conditions such as feed concentration (0.2% *w*/*v* and 0.8% *w*/*v*), inlet temperature (80 °C and 100 °C) and presence/absence of L-leucine (0% and 10%). The prepared dry powder was in the size range of 1–5 μm and amorphous in nature with wrinkle morphology. We observed a higher fine particle fraction (82.5 ± 1.4%) in high feed concentration (0.8% *w*/*v*), high inlet temperature (100 °C) and the presence of L-leucine (10% *w*/*w*). The stability study conducted for 28 days confirmed that the spray-dried powder was stable at 25 ± 2 °C/<15% RH and 25 ± 2 °C/ 53% RH. Interestingly, the ivermectin dry powder formulation inhibited SARS-CoV-2 replication in vitro with a potency similar to ivermectin solution (EC_50_ values of 15.8 µM and 14.1 µM, respectively), with a comparable cell toxicity profile in Calu-3 cells. In summary, we were able to manipulate the spray-drying conditions to develop an effective ivermectin inhalable dry powder. Ongoing studies based on this system will allow the development of novel formulations based on single or combinations of drugs that could be used to inhibit SARS-CoV-2 replication in the respiratory tract.

## 1. Introduction

SARS-CoV-2, the causative viral agent of COVID-19, quickly spread across the whole world, and as of 16 May 2022, the number of confirmed cases was over 517 million, with more than 6.2 million deaths, according to the World Health Organization (WHO) report [1]. Currently, available vaccines for COVID-19 are building immunity against SARS-CoV-2 and reducing virus transmission. However, it is uncertain how long this immunity lasts [2]. In that scenario, even vaccinated people may be able to transmit the virus, infecting other people [2]. Moreover, recent studies seem to suggest that not only the efficacy of available vaccines may vary against different variants of SARS-CoV-2 [3], but the immune response generated by the vaccines may also wane over time [4,5,6]. Therefore, it is fair to assume that SARS-CoV-2 will continue to circulate for many years to come [7]. For that reason, in addition to vaccines, there is a clear need to develop treatments for COVID-19 patients as a direct-acting or as adjunct therapy [2]. Currently, available therapeutics for this disease are mainly given based on previous experiences of treating other viral diseases or the effectiveness of drugs conducted in cell line studies [8,9].

SARS-CoV-2 enters the body through inhalation and gradually affects the nasal area as well as the lungs [10,11]. In the later phase, it spreads to the whole body from the lungs. Therefore, the lungs are the primary site of this viral infection. In the lungs, the virus binds with the ACE-2 receptor of epithelial cells, which could lead to a cytokine storm characterized by increased levels of several interleukins such as IL-6 and IL-8 [12]. This cytokine storm is the generator of a series of complications such as endotheliopathy, pulmonary embolism, acute respiratory distress syndrome and so on [11,13,14,15,16,17].

Ivermectin was approved by the U.S. FDA as an anti-parasitic drug in 1998, and the WHO included it in a List of Essential Medicines showing effectiveness against different RNA and DNA viruses [18]. Early studies during the initial phase of the COVID-19 pandemic, ivermectin showed a promising antiviral effect against SARS-CoV-2 in cell culture, using Vero/hSLAM cells [19] while a placebo-controlled study (randomized and double-blind) based on 72 COVID-19 patients in Bangladesh showed that ivermectin could shorten the illness [20]. Although the exact mechanism of action of ivermectin against SARS-CoV-2 is still unknown, it was suggested that it might prevent the binding of SARS-CoV-2 proteins to Imp α/β1, blocking the traffic of viral proteins to the nucleus [19]. Other studies associated ivermectin with a reduction in the cytokine response [21]. Despite the initial excitement, several clinical trials showed limited efficacy of ivermectin against COVID-19, leading to questioning our ability to use ivermectin at the right concentration in humans. For example, in one randomized trial of over 400 patients, the oral administration of ivermectin was found non-significant in the symptom’s recovery time [22]. In another randomized trial of over 3515 patients, orally administered ivermectin failed to lower the extent of medical admission due to the progression of COVID-19 or lower the chance of adverse effects significantly [23]. The efficacy study of ivermectin conducted over 490 patients in Malaysia found that orally administered ivermectin failed to prevent the disease progression from mild to severe conditions [24]. In fact, the pharmacokinetic modeling-based tissue distribution data on cattle concluded that even 10 times the approved oral dose of poorly absorbed ivermectin could not be sufficient to inhibit SARS-CoV-2 replication in vivo [25,26]. Therefore, it is possible that ivermectin concentrations shown to inhibit SARS-CoV-2 replication in vitro can be only safely achieved as an inhaled therapy.

Respiratory drug delivery is regarded as one of the most logical ways of treating COVID-19 infection, as it can ensure higher drug concentration at lower doses with minimum side effects. In the specific case of ivermectin and its potential use against SARS-CoV-2 infection, delivering the drug directly to the lung would bypass hepatic metabolism, ensuring high drug concentration in the lung with less systemic side effects [27,28]. A pilot study conducted on a rat model with nebulized ivermectin, with ethanol as a vehicle, showed no histological anomalies in the lung at relatively high doses of 110 to 140 mg/kg [29]. Unfortunately, nebulization may spread the highly transmissible SARS-CoV-2 [30], while using organic solvents as a vehicle to deliver poorly water-soluble drugs through nebulization could lead to potential toxicities [31]. On the other hand, inhalable dry powder formulations using spray-drying technology offer several advantages compared to nebulizers in terms of stability and administration [32]. In this study, we used ivermectin to develop a spray-dried powder by manipulating spray drying conditions such as feed concentration and inlet temperature using a factorial design. Our initial formulation contained the drug alone or with the use of L-leucine as an excipient. The physicochemical properties, in vitro aerosolization and stability of prepared dry powder were investigated, and the formulation was used to inhibit SARS-CoV-2 replication in Calu-3 cells.

## 2. Materials and Methods

### 2.1. Materials

Ivermectin (purity 97.1%) was a generous gift from Hovione Pharma Science Limited (Estrada Coronel Nicolau de Mesquita Taipa, Macau, China). L-leucine was purchased from Hangzhou Dayangchem Co., Ltd., Hangzhou, China. All the organic solvents (acetonitrile, absolute ethanol and methanol) were of High-Performance Liquid Chromatography (HPLC) grade and purchased from Merck, Darmstadt, Germany. Purified water was collected from the inbuilt Millipore continental water system (Millipore Corporation, Burlington, MA, USA).

### 2.2. Inhalable Size Ivermectin Dry Powder Preparation

Inhalable size ivermectin dry powder was prepared by the co-solvent system (90% ethanol: 10% water), using a Buchi B-290 Mini Spray-Dryer with different feed concentrations, inlet temperature and in the presence/absence of L-leucine, utilizing a 2^3^ factorial design (Table 1 and Table 2).

Each prepared dry powder was collected from the spray dryer, and the yield (%) was calculated using the below formula. The powder was then kept in a glass vial (screw-capped) and stored in a desiccator at room temperature until use.
(1)$$\mathrm{Yield}=\frac{\mathrm{Obtained}\;\mathrm{powder}\;\mathrm{weight}\;\left(\mathrm{mg}\right)}{\mathrm{Total}\;\mathrm{powder}\;\mathrm{used}\;\mathrm{in}\;\mathrm{the}\;\mathrm{formulation}\;\left(\mathrm{mg}\right)}\times 100\%$$

### 2.3. Drug Content

The drug content of prepared dry powder was assessed by High-Performance Liquid Chromatography (HPLC) using the method described in Section 2.9. Approximately 5–10 mg of prepared dry powder was dissolved in 100 mL ethanol and quantified in HPLC. The tests were performed in triplicate for each formulation.

### 2.4. Water Content Determination

The water content of ivermectin raw material and prepared ivermectin dry powders was assessed by thermogravimetric analysis with a TGA Q50 analyzer (TA Instruments, New Castle, DE, USA). Around 2–5 mg of sample (raw material or dry powder formulations) was taken to a TGA pan. The samples were then heated at a constant temperature increment (10 °C/min) from ambient temperature to 120 °C. The water content was determined based on the weight loss of the samples using the inbuilt software (TRIOS) of the instrument. The tests were performed twice for each formulation.

### 2.5. Powder Morphology and Particle Size

The powder morphology of prepared formulations was investigated by scanning electron microscopy (Carl Zeiss Inc., Oberkochen, Germany). Briefly, 5 mg powder of each sample was spread on a carbon adhesive tape and then coated with gold/palladium alloy by carbon coater (Quorum Technologies Ltd., Sussex, UK). The images of powder samples were captured at an accelerating voltage (5 kV). The diameters of at least 300 particles for each sample were measured to determine the average geometric diameter by an image analyzing software ImageJ (National Institutes of Health, Bethesda, MD, USA).

### 2.6. Powder Crystallinity Nature

The crystallinity or amorphicity of supplied ivermectin raw material, L-leucine, and prepared powder formulations was investigated by examining powder X-ray diffractions (PXRD) obtained by an X-ray diffractometer (Malvern Panalytical, Malvern, UK). All the samples were analyzed over the 2θ range of 5–35° (6°/min). Through X’Pert Data Collector (Malvern Panalytical), all the data were collected and analyzed by the software HighScore suite (Malvern Panalytical).

### 2.7. Drug–Excipient Interaction

The drug–excipient interaction was assessed by an ATR-FTIR instrument (Galdi ATR and Varian Inc., Palo Alto, CA, USA). Raw material, excipient and prepared dry powders were placed on the crystal plate of the instrument, and 64 scans were collected with a 4 cm^−1^ spectral resolution over 500–4000 cm^−1^. The obtained data were analyzed by the inbuilt software of the instrument (Varian, Palo Alto, CA, USA).

### 2.8. In Vitro Aerosolization Property

The in vitro aerosolization property of prepared ivermectin dry powder was investigated by a Next Generation Impactor (NGI) (CopleyScientific Limited, Nottingham, UK). Briefly, the airflow for powder dispersion was set to 100 L/min using a vacuum pump, critical flow controller and flow meter (Copley, UK). In order to minimize unwanted particle bounce, silicone oil (viscosity: 10^−5^ m^2^/s) was applied on NGI plates, and this mimics the thin fluid lining present in the lung. Approximately 20 mg of powder was loaded in a size 3 hard gelatin capsule and activated by an aerolizer (Foradil aerolizer, Novartis Pharmaceutical Ltd., London, UK). Then the activated capsule containing powder was aerosolized with a flow rate of 100 L/min for 2.4 s. The pressure drop across the device is ~4 kPa at 100 L/min [33,34]. These parameters with an aerolizer are well documented by researchers as this mimics the usually forced inhalation capacity of a standard-sized human [35,36]. All the deposited powders in different stages of NGI (S1–S7) and Micro-Orifice Collector (MOC), mouthpiece and inhaler containing a capsule were collected by dissolving powders in ethanol. The cut-off size of NGI stage 2 at 100 L/min is 3.42 μm [36]. The HPLC method was used to quantify the amount of dry powders. Each sample was quantified in triplicate.

The recovered dose (RD), emitted dose (ED), fine particle fraction (FPF) and fine particle dose (FPD) were calculated. The RD is the total amount of drugs obtained from the aerolizer (device), capsules, mouthpiece and different stages of the NGI. ED is the amount of drug that is actually exerted from the aerolizer (device) and deposited in the mouthpiece and different stages of NGI. The %FPF is the FPD presented as corresponding to ED. The FPD is the amount of deposited drug in S2–MOC.

### 2.9. HPLC Analysis for Drug Quantification

Ivermectin was quantified by a reverse phase isocratic HPLC (Agilent, Santa Clara, CA, USA) method. A fusion column (C18 Synergi Fusion Column, 250 mm × 4.6 mm, 5 μm, Phenomenex, Torrance, CA, USA) was used. The mobile phase composition was acetonitrile, methanol and water (60:30:10) at a flow rate of 1.7 mL/min with an oven temperature of 30 °C. The injection volume for each sample was 20 μL and detected at 246 nm wavelength with a 12 min total run time. The retention time was ~8 min. The calibration curves were linear (*R*^2^ > 0.999) over the concentration range of 1–100 μg/mL with the limit of detection (LOD) and limit of quantitation (LOQ) of 0.17 μg/mL and 0.53 μg/mL, respectively. The separately prepared quality control samples (5, 40 and 80 μg/mL) showed repeatability and reproducibility within the acceptable range (%Bias: ≤15; %Coefficient of variation: ≤15).

### 2.10. Stability Study

The spray-dried powder was stored in two different conditions (25 ± 2 °C/15% RH and 25 ± 2 °C/53% RH) for 28 days to investigate the impacts of relative humidity. The powder was distributed evenly over an open petri dish. After 28 days, the in vitro aerosolization and physicochemical properties (powder size, morphology, nature and drug content) of the dry powder were evaluated as per the methods described earlier.

### 2.11. In Vitro Permeation Testing of Ivermectin Dry Powder

The in vitro permeation (dissolution followed by diffusion through the membrane) profile of prepared ivermectin dry powder was assessed according to the method described by Eedara et al. with modifications [37]. For the in vitro permeation test, respirable-size ivermectin dry powder was collected from a modified Twin Stage Impinger (mTSI). Ivermectin dry powder formulation (~8 mg) was filled in a gelatin capsule (Size #3) and dispersed into the mTSI by an aerolizer with an airflow rate of 60 L/min for 4 s. The custom-made lower impingement chamber with the screw cap bottom eases the respirable-size dry powder collection onto the glass coverslip. The coverslip edges were covered with a passe-partout to collect the deposited dry powder from the uniform area of the coverslips. The respirable size powder deposited on the coverslips was collected by opening the lower impingement chamber.

The in vitro permeation test was carried out using a low-volume custom-built dissolution apparatus with a flow perfusion cell attached to a 100 DM syringe pump (Teledyne ISCO, Lincoln, NE, USA) (Figure 1). A dialysis membrane (Molecular weight cut off = 12,400 Da, Sigma Aldrich, Burlington, MA, USA) soaked in PBS (pH 7.4) was fixed in the flow perfusion cell. The thickness of the dialysis membrane was ~60 µm. The temperature of the perfusate medium filled into the reservoir cylinder of the pump was maintained at 37 ± 0.5 °C by water circulation through the temperature control jacket by a heating circulator (Grant Instruments Ltd., Shepreth, UK). The dissolution medium consisted of polyethylene oxide (1.5% *w*/*w*) and Curosurf^®^ (0.4% *w*/*w*) in phosphate-buffered saline (PBS, pH 7.4), and 50 μL of it was used for the experiment. Briefly, in a glass vial, 9810 mg of PBS (pH 7.4), 40 mg of Curosurf^®^ and 150 mg of polyethylene oxide were weighed and stirred overnight to prepare the dissolution medium. The perfusate was Tween 80 (0.2% *w*/*v*) in PBS solution, and the flow rate of the perfusate was set at 0.05 mL/min. The powder deposited on the coverslip was brought into contact with the dissolution medium on the membrane. As it was an open mode system, a fixed temperature could not be maintained at the surface of the membrane. However, the temperature was measured to be 31 ± 4 °C throughout the experiment. The samples were collected over 30 h at different time points, i.e., 15 min, 30 min, 1 h, 1 h 30 min, 2 h, 4 h, 8 h, 12 h, 24 h and 30 h. Finally, the quantification of the drug in the samples was conducted by the developed HPLC method.

### 2.12. Cellular Toxicity Assay

The potential cytotoxic effect of ivermectin, as well as the ivermectin dry powder and L-leucine alone, was determined by quantifying both cell viability and cellular proliferation in different cell lines using the XTT [2,3-Bis-(2-Methoxy-4-Nitro-5-Sulfophenyl)-2*H*-Tetrazolium-5-Carboxanilide] colorimetric method in the Cell Proliferation Kit II (Merck Sigma-Aldrich). Briefly, 2 × 10^4^ Calu-3 cells/well were seeded in a 96-well plate and incubated at 37 °C, 5% CO_2_ overnight. The cell culture medium (Minimum Essential Medium Eagle and 10% Fetal Bovine Serum) was replaced with fresh media containing corresponding compounds at eight different concentrations, starting at 100 µM and prepared by serial 1:2 dilutions with the cell culture medium, and incubated at 37 °C, 5% CO_2_ for 48 h. The stock solutions were prepared by dissolving these compounds individually in dimethyl sulfoxide (DMSO). The cell culture medium was then carefully removed, and cells were washed twice with 1× PBS. Fifty microliters of fresh XTT labeling reagent (Cell Proliferation Kit II, Merck Sigma-Aldrich) were added to each well, incubated at 37 °C, 5% CO_2_ for 4 h, and absorbance quantified at 570 nm in a Varioskan LUX multimode microplate reader (Thermo Fisher Scientific, Waltham, MA, USA). The 50% cytotoxic concentration (CC_50_), defined as the concentration of compound that caused cell death or inhibited cell proliferation by 50%, was calculated using GraphPad Prism v.9.2.0 (GraphPad Software, La Jolla, CA, USA). The experiments were conducted in triplicate.

### 2.13. Drug Susceptibility Assay Based on Replication-Competent SARS-CoV-2

The susceptibility of the SARS-CoV-2 isolate hCoV-19/New Zealand/NZ1_patient/2020 [39] to (i) ivermectin, (ii) the ivermectin dry powder with L-leucine (F8) and (iii) L-leucine was evaluated in Calu-3 cells. Serial dilutions spanning empirically determined ranges of each compound were added in triplicate to 96-well plates containing Calu-3 cells (20,000 cells/well) and incubated at 37 °C, 5% CO_2_ for two hours. Cells were then infected with SARS-CoV-2 at an MOI of 0.005 IU/cell for one hour at 37 °C, 5% CO_2_. Virus inoculum was removed, cells washed twice and the complete medium with the corresponding agent dilution replenished. SARS-CoV-2 replication was quantified 72 h post-infection by CPE and a cell protection assay based on the Pierce™ BCA Protein Assay Kit (Thermo Fisher Scientific), as described [39]. Concentrations of the compounds required to inhibit SARS-CoV-2 replication by 50% (EC_50_) were calculated by plotting the percent inhibition of virus replication versus log_10_ drug concentration and fitting the inhibition curves to the data using nonlinear regression analysis (GraphPad Prism v.9.3.1, GraphPad Software).

### 2.14. Statistical Analysis

The statistical analyses were conducted with Instat Graph Pad Prism (version 5, San Diego, CA, USA), Minitab (version 16, State College, PA, USA) and Microsoft Excel. One-way ANOVA for in vitro aerosolization was conducted using Tukey’s test as a post hoc test (*p* < 0.05). The analysed data are expressed as mean ± standard deviation.

## 3. Results and Discussion

### 3.1. Selection of Spray-Drying Process Parameters and Excipient

The inlet temperature of the spray-dryer can affect the physical properties of the active ingredient, which in consequence affect the overall properties of the powder. For example, the powder can be fused if the inlet temperature is high. Again, the moisture content and yield are highly dependent on the inlet temperature [40]. On the other hand, the feed concentration can affect the particle size, in vitro aerosolization properties and moisture content of the dry powder. For pulmonary drug delivery, particle size with an aerodynamic diameter between 1 and 5 μm is desirable [35]. The feed concentration showed a greater influence on the particle size of spray-dried powders compared to other spray drying factors such as solvent system (aqueous/organic/co-solvent), feed rate, atomizing airflow, drying air flow rate and moisture in drying air [40]. L-leucine is generally regarded as a safe (GRAS) amino acid for pulmonary delivery and has aerosolization enhancement properties due to the surface activity and low water solubility [36,41,42,43]. L-leucine was used in this study as it is not bronchospasm or cough-inducing like the sugar-based excipients (e.g., mannitol). The generation of cough may spread the virus from the individual through droplet generation, which is not desired in a contagious disease such as COVID-19 [44]. L-leucine can protect the active ingredient from moisture by forming a barrier on the surface through crystallization, which prevents water absorption [45,46]. The formulations were spray-dried using fixed aspiration (80%), drying gas flow rate (670 L/h), pump feeding rate (1 mL/min) and nozzle diameter (0.7 mm). Ethanol and water co-solvent (90:10) was chosen as ivermectin is practically insoluble in water but soluble in organic solvents (e.g., ethanol, acetonitrile, etc.), and ethanol is in The International Council for Harmonisation of Technical Requirements for Pharmaceuticals for Human Use (ICH) class 3, which means low toxic potential solvent group [47,48]. On the other hand, L-leucine is soluble in water. The melting band of ivermectin was found at 155–157 °C, and the boiling point of ethanol is ~78 °C [47]. In our preliminary study, we observed that, at an inlet temperature of 140 °C or 120 °C, the supplied ivermectin fused when spray-dried (Appendix A). Therefore, 100 °C and 80 °C were selected as the high and low values of inlet temperatures. We used a low amount of L-leucine (10% *w*/*w*) to ensure more drug loading. The higher feed concentration was selected based on the solubility of ivermectin and L-leucine in the co-solvent.

### 3.2. Process Yield, Drug Content and Water Content

The yield, drug content and water content of spray-dried formulations are tabulated in Table 3. The batch size was 500 mL for the formulations with 0.2% (*w*/*v*) feed concentration (F-1, F-3, F-5, F-7) and 125 mL for the formulations with 0.8% (*w*/*v*) feed concentration (F-2, F-4, F-6, F-8). The yield of all the spray-dried formulations was over 42%, with the maximum yield of ~60% for F-8, which was prepared in high feed concentration (0.8% *w*/*v*), a high inlet temperature (100 °C) and the presence of L-leucine (10% *w*/*w*). The yield for spray-dried formulation generally lies between 20 and 50% [32], and various parameters such as inlet temperature, feed concentration and the presence of excipients can influence the yield. For example, a higher yield can be obtained by increasing the inlet temperature and the feed concentration [40]. Additionally, the addition of L-leucine can also increase the yield of spray-dried powder [49,50]. The yield for dry powder prepared at 100 °C was 50–60% compared to the yield of 42–47% prepared at 80 °C. The reason behind obtaining a higher yield at high inlet temperature is that at high inlet temperature, the powder particles have low moisture content. As they have less moisture content, these powders have good flowability and are not sticky and do not remain in the spray-dryer wall [49]. However, the inlet temperature needs to be chosen in a way that does not cause any thermal degradation of the selected agent(s). The yield for dry powder prepared at 0.2% *w*/*v* was 42.5%, 43.7%, 50.7% and 51.2%, whereas the yield for dry powder prepared at 0.8% *w*/*v* was higher with the values of 46.1%, 46.9%, 58.7% and 59.9%, respectively (at same inlet temperature and presence/absence of L-leucine). The increase in feed concentration allows less solvent to evaporate, which leads to more dry powder with less moisture content [40]. Again, the high feed concentration produces larger particles because of more amounts of solid, which are easy to collect and increase the yield [40]. The higher value of feed concentration is mainly chosen in a way that the solid content is fully soluble in the solvent. Although the yield is normally lower with smaller batch size, the effect was not evident in this study, probably due to the influence of feed concentration. For example, the batch size of F-2 (125 mL) was smaller than F-1 (500 mL, but the yield of F-2 (46.1%) was larger than F-1 (42.5%) as the feed concentration of F-2 (0.8% *w*/*v*) is higher than F-1 (0.2% *w*/*v*). Additionally, the L-leucine-containing formulations (at the same inlet temperature and feed concentration) have a higher yield compared to the formulations without L-Leucine (42.5% vs. 43.7%), (46.1% vs. 46.9%), (50.7% vs. 51.2%) and (58.7% vs. 59.9%). L-leucine may have changed the cohesive force of the prepared dry powder and reduced the powder deposition on the spray dryer wall and more powder deposition in the collector [50].

The drug content of the spray-dried powders was found to be 95% to 105% for all the formulations. The drug content of three formulations was higher than 100% (101.9%, 102.7% and 103.6% for F5, F-7 and F-8). The slightly higher drug content could be due to the analytical error or non-homogeneous distribution of the drug and excipient. The acceptance limits of drug content for inhaled formulations are 75–125% and 85–115%, according to British Pharmacopoeia and the United States Pharmacopoeia, although the industry follows even more stringent limits, often 90–110% [51,52,53,54,55]. However, the drug content of all the formulations in this study was within ±5%, the above-mentioned acceptable limit. All the ivermectin dry powder formulations have less than 1% water content. One possible reason could be the hydrophobic nature of ivermectin and L-leucine. Another reason could be the use of high organic solvent and high inlet temperature (80 °C or 100 °C), which allows producing more dry particles. Moisture content below 5% *w*/*w* is preferred as more than that may impair stability and in vitro aerosolization [32,56]. The low moisture content is necessary for good flowability and limited agglomeration and stickiness [32].

### 3.3. Powder Morphology and Particle Size

The spray-dried ivermectin powders showed wrinkle morphology with irregular structures (Figure 2). This irregular structure and wrinkle morphology can improve in vitro aerosolization property [57]. One possible reason for this morphology can be the co-solvent (ethanol: water) system [58]. In this study, the morphology of prepared ivermectin dry powder was similar in the presence/absence of L-leucine. It is consistent with the report that L-leucine does not have much impact on the morphology when co-spray-dried with hydrophobic drugs [59]. This can be relatable with our findings as ivermectin is a hydrophobic drug.

Ivermectin raw material was tabular in shape and higher than the inhalable size range (Figure 2 and Appendix A) of 1–5 μm. Particles with an aerodynamic diameter less than 1 μm are generally exhaled, and particles with an aerodynamic diameter larger than 5 μm are deposited in the upper respiratory tract (extra-thoracic region) [35]. The particle size and median diameters of the size distribution (D_50_) play a key role in the aerodynamic diameter properties [49,60]. The median size distribution (D_50_) indicates that half of the total particle size is below the number, and the other half is above that number. The D_50_ values for F-1. F-2, F-3, F-4, F-5, F-6, F-7 and F-8 formulations were 1.7, 2.2, 1.9, 1.8, 1.9, 2.2, 1.9 and 2.1 μm, respectively (Table 3). The geometric diameters/particle size of prepared ivermectin dry powders were within 1–5 μm for all the formulations (Table 3). The feed concentration and L-leucine had impacts on the particle size of the prepared ivermectin dry powder, and no such impact was found for inlet temperature. For example, the average particle size was found higher when the feed concentration increased from 0.2% *w*/*v* to 0.8% *w*/*v* (e.g., 2.1 to 2.5 μm***, 1.9 to 2.4 μm***, 2.2 to 2.6 μm* and 1.9 to 2.3 μm*) as the solid amount while producing droplets is more at higher feed concentration compared to low feed concentration [40]. The impact of feed concentration on the average particle size was significant (*p* < 0.05). In contrast, the addition of L-leucine in the formulation produced smaller particles. For example, 2.1 to 1.9 μm, 2.5 to 2.4 μm, 2.2 to 1.9 μm and 2.6 to 2.3 μm. The reason is that as it is a hydrophobic amino acid, during the fast and quick drying process, it generally migrates to the powder surface and affects the particle size [50]. This impact of L-leucine was not statistically significant.

### 3.4. Powder Nature

The supplied ivermectin, L-leucine and spray-dried L-leucine were crystalline in nature, revealed by X-ray diffractograms (XRD) peaks (Figure 3) and also similar to other reports [45,50]. In contrast, no peak of all the prepared spray-dried powders confirmed the amorphous nature of all the formulations (Figure 3). In spray-drying technology, the powder is generally obtained in an amorphous form [61].

### 3.5. Drug–Excipient Interaction

The spectra obtained from ATR-FTIR for ivermectin raw material and all the formulations confirmed no interaction in prepared ivermectin dry powders during spray-drying, as similar peaks were observed in the same position for raw ivermectin and spray-dried ivermectin powders (Figure 4). The major peaks of ivermectin raw material were observed at 3480 cm^−1^ (O-H stretchings), 2965 cm^−1^ (C-H stretchings), 1735 cm^−1^ (C=O) and 1341–1000 cm^−1^ (symmetric and asymmetric C-O-C stretchings), which are similar to the published report [47]. The formulations F-3, F-4, F-7 and F-8 showed a peak at 1529 cm^−1,^ which is identical to the peak of L-leucine (flat for ivermectin formulations without L-leucine, which are F-1, F-2, F-5, F-6) meaning the presence of L-leucine in these formulations. However, the peak intensity was decreased in the formulations containing L-leucine compared to the L-Leucine free formulations. Our group previously reported that when kanamycin was spray-dried with L-leucine, the formulations showed similar peaks position to kanamycin, but the peak intensity was decreased in formulations containing L-leucine [52].

### 3.6. In Vitro Aerosolization

The in vitro aerosolization behavior of prepared formulations was investigated and shown in Figure 5 and tabulated in Appendix A. The recovery of all the formulations was within a range of 93–105% (Appendix A). The average ED and FPF for all the formulations were over 70% and 75%, respectively. Among all formulations, F-8 showed the highest FPF of 82.5 ± 1.4%, with an ED of 77.9 ± 0.5%. As per the current literature, the obtained ED and FPF are acceptable [62,63].

The in vitro aerosolization is affected by the particle size and morphology. In this study, in vitro aerosolization was influenced primarily by feed concentration (Figure 6) and not by the addition of L-leucine or variation in inlet temperatures (Appendix A). As discussed earlier, the feed concentration and L-leucine can both alter the particle size and morphology. However, the inlet temperature did not have an impact on the particle size and the morphology of prepared ivermectin dry powder in this study.

An increasing feed concentration from 0.2% *w*/*v* to 0.8% *w*/*v* showed higher ED and FPF. For example, the average ED and FPF of ivermectin dry powder prepared at 0.2% *w*/*v* and 80 °C without L-leucine were 71.5% and 76.3%, respectively, whereas the average ED and FPF values were 76.1% and 79.9%, respectively, when prepared at 0.8% *w*/*v*. Similarly, when the dry powder was prepared at 100 °C without L-leucine, the average ED increased from 75.4% to 79.4%, and FPF increased from 75.4% to 80.2% for 0.8% *w*/*v* feed concentration.

The presence of L-leucine improved the FPF but only up to 1–2% in this study. For example, dry powders prepared at 0.2% *w*/*v* and 80 °C without L-leucine having an average ED of 71.5% with an average FPF of 76.3%, but with L-leucine, the ED and FPF were 73.8% and 77.9%, respectively. In the case of 100 °C, the average ED and FPF for L-leucine-containing formulations were 79.4% and 80.2%, respectively, which were higher than the L-leucine-free formulations (75.4% of ED and 75.4% of FPF). L-leucine has the in vitro aerosolization enhancement property [52,64], but it may not have much impact on the fine particle fraction of a hydrophobic drug. The reason may be the surface of the composite particles is enriched with hydrophobic drugs [59].

### 3.7. Stability Study

The stability study was carried out for 28 days under two different conditions (25 °C/<15% RH and 25 °C/53% RH) for four formulations (F-5, F-6, F-7 and F-8) among eight formulations (Table 2) to find out how the storage humidity affect the in vitro aerosolization of the formulations. The impact of inlet temperature was not prominent enough in the particle size and in vitro aerosolization; thus, all the formulations prepared using the inlet temperature of 80 °C were excluded for the stability study (F1, F2, F3 and F4). Moreover, the highest %ED and %FPF was obtained for F-6 and F-8, respectively, which were prepared at 100 °C. The selected formulations for the stability study were prepared at two different feed concentrations (0.2% *w*/*v* and 0.8% *w*/*v*) and in the presence or absence of L-leucine.

The ED and FPF after the stability study are shown in Appendix A and Appendix A. A comparison between the ED and FPF before (Day 0) and after storage (Day 28) is shown in Figure 7. The average FPF of F-5, F-6, F-7 and F-8 were 80.4%, 85.5%, 77.6% and 83.6%, respectively, at 25 °C/<15% RH, whereas the values were 82.2%, 86.6%, 79.1% and 86.9% at 25 °C/53% RH. Therefore, the FPF was increased in both conditions compared to before the storage of the formulations. In fact, the FPF for F-6 significantly (*p* < 0.05) increased after storage at both 25 °C/<15% RH and 25 °C/<53% RH for 28 days. The FPF increase was significant for F-5 (*p* < 0.05) only after storage at 25 °C/<53% RH for 28 days. It was reported that the FPF of powder particles could be increased in the presence of relative humidity, and it varies. Studies also revealed that the impact of relative humidity is mainly drug-specific. For example, the impact of relative humidity on the aerosolization behavior of Inhalac 230 (lactose), salbutamol sulfate and their binary mixtures showed that at 25 °C/20% RH condition, the FPF was 19% higher than before storage value and at 25 °C/43% RH condition the FPF enhanced 24% compared to the before storage FPF [56]. In another study, the FPF of di-sodium cromoglycate reduced when the relative humidity was increased from 15% to 75%, whereas the FPF of triamcinolone acetonide increased in the same condition [65]. The possible reason for increasing aerosolization could be the reduction in inter-particle adhesion during storage [66]. The electrostatic charge of dry powder may affect aerosol performance. This electrostatic charge decay varies by the drugs and excipients’ properties along with the manufacturing technique [65]. Thus the decrease in change could be one possible reason that the F-5 and F-6 showed increases in aerosol performance after storage but not F-7 and F-8. The F-7 and F-8 were L-leucine-containing formulations, whereas F-5 and F-6 were L-leucine-free formulations. The electrostatic charge decay may be more in F-5 and F-6, resulting in increased aerosol performance than in F-7 and F-8. However, aerosolization may decrease when the moisture content in the formulation increases. The water content of spray-dried ivermectin formulations was below 1% even after storage at 25 °C/<53% RH (Appendix A); this moisture content might be very low to have a negative impact on aerosolization. This could be due to the hydrophobic nature of ivermectin and L-leucine.

The wrinkle shape of spray-dried powders remained unchanged in both conditions after 28 days (Figure 8). The nature of spray-dried formulations also remained amorphous (Appendix A). The drug content for all the formulations was within 90–100% (data not shown here), which indicates the chemical stability of prepared dry powders for the short term. The ATR-FTIR study confirmed no interaction after the stability study (Appendix A). The peaks were similar before and after the storage of the formulations.

### 3.8. In Vitro Permeation Profile of Ivermectin Dry Powder

The in vitro permeation (dissolution followed by diffusion) of F-8 was carried out because it showed the highest ED and FPF (Figure 5) among all the formulations. By using this apparatus, the dissolution rate of ivermectin could not be directly measured; rather, the drug permeated through the membrane after it had dissolved in the dissolution medium was measured. The cumulative percent of ivermectin permeated into the perfusate over time is shown in Figure 9. The amount of respirable ivermectin dry powder deposited on one coverslip was 90 ± 10 µg, which was used for the in vitro permeation study. In the in vitro permeation test, ~68% of ivermectin was permeated in 30 h. The solubility of ivermectin in water is very low (~4 mg/L) [67], and the lipophilic nature of ivermectin [68] resulted in the low cumulative percent of ivermectin in the perfusate.

Furthermore, the cumulative percent of ivermectin in perfusate linearly increased with time in this study. It is probably due to the saturation of the 50 μL dissolution medium by the drug tested for dissolution. A similar trend was also observed by Hassoun et al. for a poorly water-soluble drug, fluticasone propionate, when tested using the DissolvIt system [69]. This is an interesting behavior for dissolution in a low volume of dissolution medium simulating lung conditions as opposed to dissolution study using large volumes of dissolution medium for orally administered drugs. Further studies will be required to elucidate the dissolution behavior and interplay of contributing factors.

### 3.9. In Vitro Cellular Toxicity and Anti-SARS-CoV-2 Activity of the Ivermectin Dry Powder Formulation

We first compared the effect of ivermectin and L-leucine alone with the ivermectin dry powder in cell viability and proliferation. While no cytotoxicity was observed when Calu-3 cells were exposed to L-leucine (CC_50_ value > 100 µM), both ivermectin alone and in a dry powder formulation with L-leucine showed cytotoxic effect at high concentrations (CC_50_ values of 28.3 and 39.1 µM, respectively, Figure 10A). Similar CC_50_ values were described for ivermectin with Calu-3 (22 µM) and other cell lines, such as VeroE6 (8 µM), A549 (6.2 µM), Huh-7 (12.1 µM) [70,71]. We then tested the anti-SARS-CoV-2 activity of the ivermectin dry powder and demonstrated that the formulation and addition of L-leucine did not affect the ability of ivermectin to inhibit SARS-CoV-2 replication in Calu-3 cells, showing similar EC_50_ values for ivermectin alone (14.1 µM) or formulated with L-leucine (15.8 µM), while no antiviral effect was observed with L-leucine alone (>100 µM, Figure 10B). Previous studies described slightly lower, but comparable EC_50_ values for ivermectin in vitro, ranging from 2.4 µM to 8 µM, depending on the system (cell line and readout) used [19,70,71,72,73].

## 4. Conclusions

We were able to develop a stable inhalable ivermectin dry powder formulation while maintaining the same cellular toxicity and anti-SARS-CoV-2 activity profile of the original ivermectin. The effect of feed concentration was prominent on the in vitro aerosolization behavior compared to the inlet temperature and L-leucine incorporation. The emitted dose and fine particle fraction of prepared ivermectin dry powder were augmented after the stability study conducted on different humidity (<15% RH and 53% RH). We are now using the various spray-drying parameters described in this study to develop similar formulations based on more potent and safe antiviral drugs, including double or triple-drug combinations. Although numerous studies are currently focused on the development of an effective treatment for COVID-19, the system used in this study may allow the development of inhalable drugs as dry powder formulations, which could be used as cost-effective solutions to treat and/or prevent SARS-CoV-2 infections and other viral respiratory infections.

## Figures and Tables

**Figure 1 pharmaceutics-14-01432-f001:**
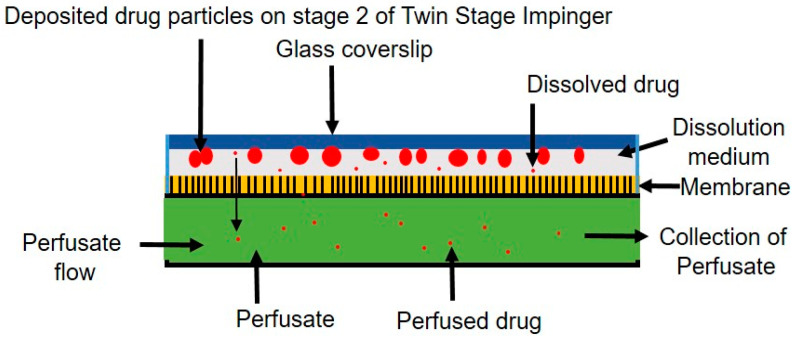
Schematic diagram of custom-built dissolution apparatus showing respirable size particles collected on cover slips dissolved in dissolution medium and became diffused through the membrane into the perfusate. Adapted with permission from Ref. [38]. 2019, Elsevier.

**Figure 2 pharmaceutics-14-01432-f002:**
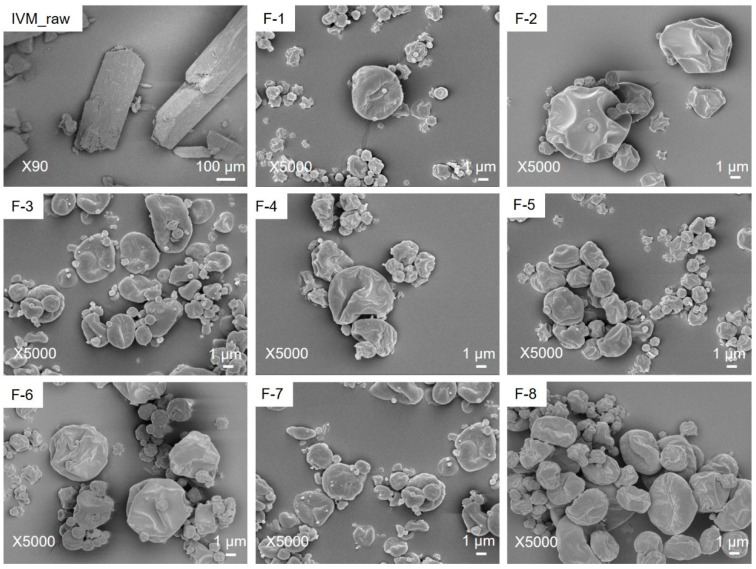
Powder morphology of ivermectin raw material (IVM_raw) and spray-dried ivermectin formulations at different conditions: (F-1) 0.2% (*w*/*v*) feed concentration, 80 °C and 0% L-leucine; (F-2) 0.8% (*w*/*v*) feed concentration, 80 °C and 0% L-leucine; (F-3) 0.2% (*w*/*v*) feed concentration, 80 °C and 10% L-leucine; (F-4) 0.8% (*w*/*v*) feed concentration, 80 °C and 10% L-leucine; (F-5) 0.2% (*w*/*v*) feed concentration, 100 °C and 0% L-leucine; (F-6) 0.8% (*w*/*v*) feed concentration, 100 °C and 0% L-leucine; (F-7) 0.2% (*w*/*v*) feed concentration, 100 °C and 10% L-leucine; (F-8) 0.8% (*w*/*v*) feed concentration, 100 °C and 10% L-leucine. Ivermectin raw material has tabular morphology, whereas all the spray dried formulations have wrinkle morphology.

**Figure 3 pharmaceutics-14-01432-f003:**
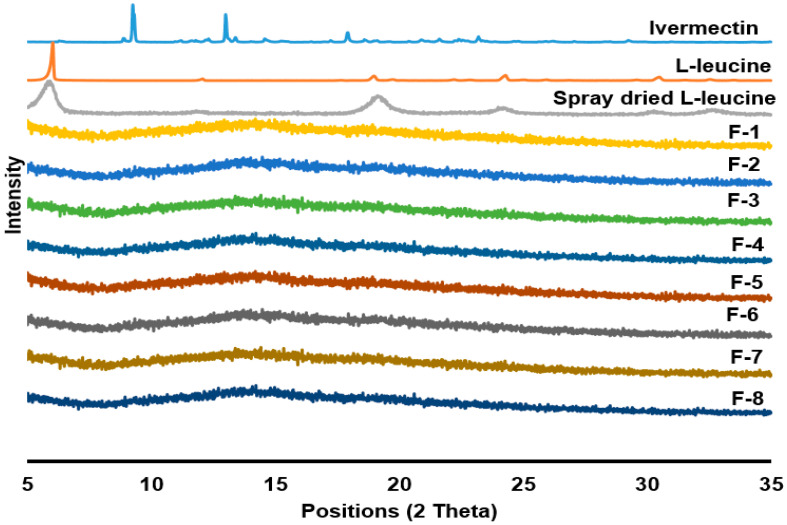
X-ray diffractograms (XRD) of supplied ivermectin, L-leucine, spray-dried L-leucine and spray-dried formulations at different conditions: (F-1) 0.2% (*w*/*v*) feed concentration, 80 °C and 0% L-leucine; (F-2) 0.8% (*w*/*v*) feed concentration, 80 °C and 0% L-leucine; (F-3) 0.2% (*w*/*v*) feed concentration, 80 °C and 10% L-leucine; (F-4) 0.8% (*w*/*v*) feed concentration, 80 °C and 10% L-leucine; (F-5) 0.2% (*w*/*v*) feed concentration, 100 °C and 0% L-leucine; (F-6) 0.8% (*w*/*v*) feed concentration, 100 °C and 0% L-leucine; (F-7) 0.2% (*w*/*v*) feed concentration, 100 °C and 10% L-leucine; (F-8) 0.8% (*w*/*v*) feed concentration, 100 °C and 10% L-leucine. Ivermectin, L-leucine and spray dried L-leucine are crystalline, which was confirmed from the XRD peaks, whereas no peaks in the formulations (F-1 to F-8) indicate the amorphous nature.

**Figure 4 pharmaceutics-14-01432-f004:**
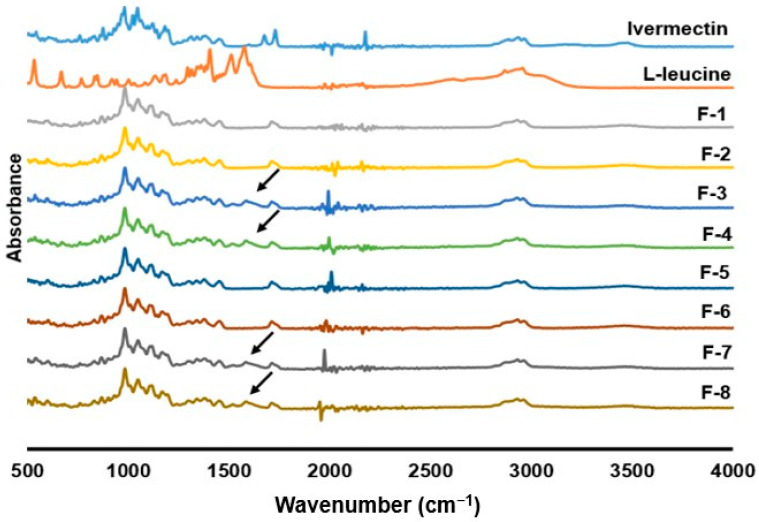
ATR-FTIR spectra of supplied ivermectin, L-leucine and spray dried formulations at different conditions: (F-1) 0.2% (*w*/*v*) feed concentration, 80 °C and 0% L-leucine; (F-2) 0.8% (*w*/*v*) feed concentration, 80 °C and 0% L-leucine; (F-3) 0.2% (*w*/*v*) feed concentration, 80 °C and 10% L-leucine; (F-4) 0.8% (*w*/*v*) feed concentration, 80 °C and 10% L-leucine; (F-5) 0.2% (*w*/*v*) feed concentration, 100 °C and 0% L-leucine; (F-6) 0.8% (*w*/*v*) feed concentration, 100 °C and 0% L-leucine; (F-7) 0.2% (*w*/*v*) feed concentration, 100 °C and 10% L-leucine; (F-8) 0.8% (*w*/*v*) feed concentration, 100 °C and 10% L-leucine (arrows indicating L-leucine peak).

**Figure 5 pharmaceutics-14-01432-f005:**
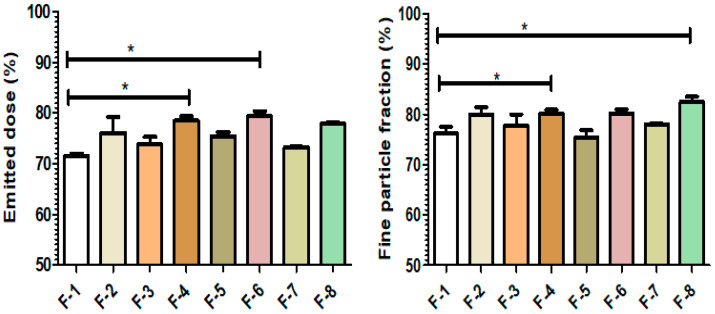
In vitro aerosolization of prepared ivermectin dry powder at different conditions: (F-1) 0.2% (*w*/*v*) feed concentration, 80 °C and 0% L-leucine; (F-2) 0.8% (*w*/*v*) feed concentration, 80 °C and 0% L-leucine; (F-3) 0.2% (*w*/*v*) feed concentration, 80 °C and 10% L-leucine; (F-4) 0.8% (*w*/*v*) feed concentration, 80 °C and 10% L-leucine; (F-5) 0.2% (*w*/*v*) feed concentration, 100 °C and 0% L-leucine; (F-6) 0.8% (*w*/*v*) feed concentration, 100 °C and 0% L-leucine; (F-7) 0.2% (*w*/*v*) feed concentration, 100 °C and 10% L-leucine; (F-8) 0.8% (*w*/*v*) feed concentration, 100 °C and 10% L-leucine (* indicating *p* < 0.05).

**Figure 6 pharmaceutics-14-01432-f006:**
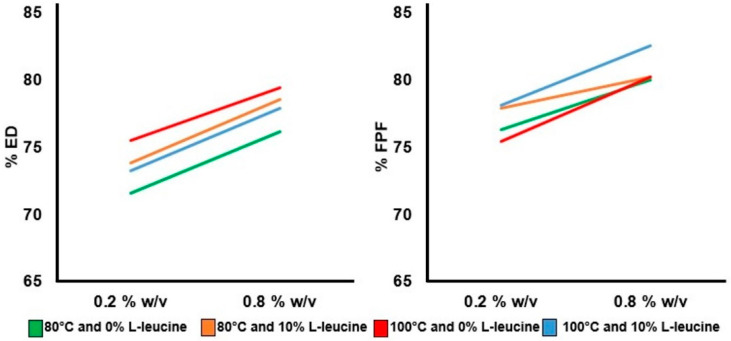
Influence of feed concentrations on emitted dose (ED) and fine particle fraction (FPF). The emitted dose and fine particle fraction were higher at high feed concentration (0.8% *w*/*v*) compared to lower feed concentration (0.2% *w*/*v*).

**Figure 7 pharmaceutics-14-01432-f007:**
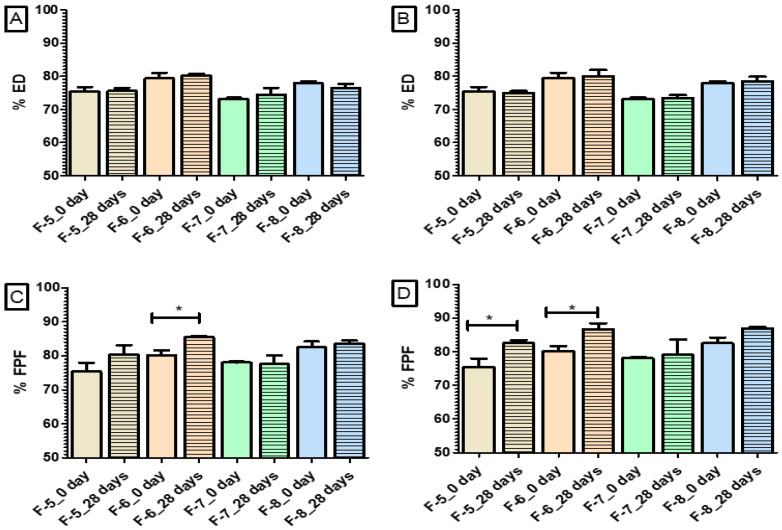
Emitted dose (ED) on: (**A**) day 0 and 28 days at 25 °C/<15% RH; (**B**) day 0 and 28 days at 25 °C/<53% RH, and fine particle fraction (FPF); (**C**) day 0 and after 28 days at 25 °C/<15% RH; (**D**) day 0 and after 28 days at 25 °C/<53% RH. The parameters of stated formulations are—(F-5) 0.2% (*w*/*v*) feed concentration, 100 °C and 0% L-leucine (F-6) 0.8% (*w*/*v*) feed concentration, 100 °C and 0% L-leucine (F-7) 0.2% (*w*/*v*) feed concentration, 100 °C and 10% L-leucine (F-8) 0.8% (*w*/*v*) feed concentration, 100 °C and 10% L-leucine (* indicating *p* < 0.05).

**Figure 8 pharmaceutics-14-01432-f008:**
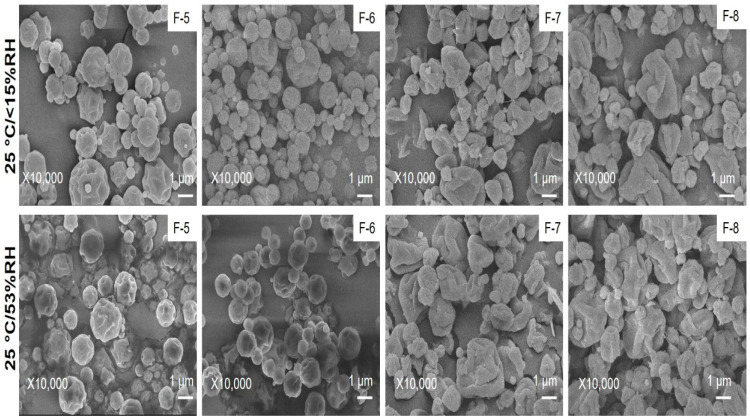
Powder morphology of spray-dried formulations at 25 °C/<15% RH and 25 °C/53% RH after 28 days prepared at different conditions: (F-5) 0.2% (*w*/*v*) feed concentration, 100 °C and 0% L-leucine; (F-6) 0.8% (*w*/*v*) feed concentration, 100 °C and 0% L-leucine; (F-7) 0.2% (*w*/*v*) feed concentration, 100 °C and 10% L-leucine; (F-8) 0.8% (*w*/*v*) feed concentration, 100 °C and 10% L-leucine. All the dry powders have wrinkle morphology.

**Figure 9 pharmaceutics-14-01432-f009:**
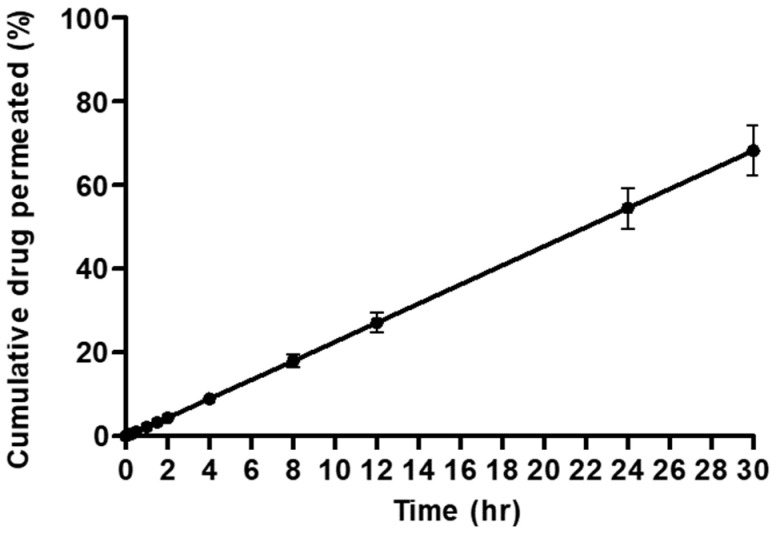
Permeation (dissolution followed by diffusion) profile of ivermectin dry powder formulation (F-8). An amount of 50 μL of dissolution medium (polyethylene oxide (1.5% *w*/*w*) and Curosurf^®^ (0.4% *w*/*w*) in phosphate-buffered saline) was used in this experiment. The flow rate of the perfusate (Tween 80 (0.2% *w*/*v*) in PBS solution) was set at 0.05 mL/min. The amount of drug loaded on the apparatus was 90 ± 10 µg.

**Figure 10 pharmaceutics-14-01432-f010:**
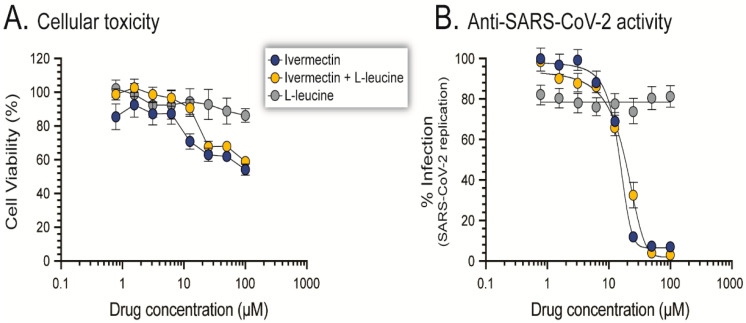
In vitro (**A**) cellular toxicity and (**B**) anti-SARS-CoV-2 activity of ivermectin raw material, L-leucine and ivermectin dry powder formulation containing L-leucine (F-8) were evaluated in Calu-3 cells as described in Materials and Methods.

**Table 1 pharmaceutics-14-01432-t001:** Factorial design conditions.

	Levels	
Factors	Low Value	High Value
Feed concentration (% *w*/*v*)	0.2	0.8
Inlet temperature (°C)	80	100
L-leucine (% *w*/*w*)	0	10

**Table 2 pharmaceutics-14-01432-t002:** The full 2^3^ factorial design for dry powder formulation preparation.

Formulation	Feed Concentration (% *w*/*v*)	Inlet Temperature (°C)	L-Leucine (% *w*/*w*)
F-1	0.2	80	0
F-2	0.8	80	0
F-3	0.2	80	10
F-4	0.8	80	10
F-5	0.2	100	0
F-6	0.8	100	0
F-7	0.2	100	10
F-8	0.8	100	10

**Table 3 pharmaceutics-14-01432-t003:** Process yield, particle size, drug and water content of ivermectin dry powder.

Formulation	Yield (%)	Size (μm)	D_50_ (μm)	Drug Content (%)	Water Content (%)
F-1	42.5	2.1 ± 1.4	1.7	98.4 ± 0.1	0.7 ± 0.2
F-2	46.1	2.5 ± 1.3	2.2	97.6 ± 0.1	0.2 ± 0.1
F-3	43.7	1.9 ± 0.7	1.9	95.9 ± 0.2	0.3 ± 0.1
F-4	46.9	2.4 ± 1.7	1.8	97.1 ± 0.2	0.3 ± 0.1
F-5	50.7	2.2 ± 1.2	1.9	101.9 ± 0.4	0.5 ± 0.1
F-6	58.7	2.6 ± 1.6	2.2	98.8 ± 0.1	0.3 ± 0.1
F-7	51.2	1.9 ± 1.0	1.9	102.7 ± 0.2	0.4 ± 0.1
F-8	59.9	2.3 ± 1.2	2.1	103.6 ± 0.5	0.4 ± 0.1

## Data Availability

Raw data can be requested by contacting the corresponding author.

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
