# Peer review of "Manipulation of Spray-Drying Conditions to Develop an Inhalable Ivermectin Dry Powder"

_pharmaceutics, 2022, doi:10.3390/pharmaceutics14071432_

Round 1
Reviewer 1 Report
Firstly, thank you to the editor and authors to give me this chance to re-review this interesting work. The manuscript after revision and adding the part of in vitro work is persuasive and its application has more chances.
Reviewer 2 Report
Attached PDF

Round 2
Reviewer 2 Report
Line 29: The word ‘original’ must be replaced with ‘ivermectin solution’ if the cell toxicity assay in section 2.12 was performed with an ivermectin solution.
Moreover in the same section, Line 287 the sentence ‘empirically determined serial dilutions of the corresponding compounds’ should be replaced with a detailed description of the solutions prepared for ivermectin, ivermectin dry powder and L-leucine alone (starting concentrations, solvent or mixture of solvents) and the dilutions made with the cell medium to treat the cells.
A Schematic Figure of the dissolution-permeation apparatus must be included to enhable the reader to follow the description in section 2.11.
An explanation of the values above 100% ‘in Table 3 must be reported in the text.
Figure 5: Have the axes been drawn on the same figure? Please check.
Author Response
A point by point response is attached.
